# Position: Don't Just "Fix it in Post": A Science of AI Must Study Training Dynamics

**Stella Biderman** [1]   **Mohammad Aflah Khan** [2]   **Niloofar Mireshghallah** [3]   **Catherine Arnett** [1]   **Fazl Barez** [4]   **Naomi Saphra** [5,6]

## Abstract

What would it mean to have a *scientific* understanding of AI? Models are not static objects: they are snapshots of time-evolving processes shaped by data, objectives, architectures, and optimization dynamics. Yet much of AI research treats models as fixed artifacts, analyzing behaviors after training rather than asking why they emerge. This position paper argues that a science of AI must move beyond post-hoc fixes and study the training dynamics that produce model behavior. Such a science should support progressively stronger forms of understanding: predicting outcomes from early training signals, intervening when trajectories go wrong, and ultimately designing training procedures that more reliably produce desired properties. Scaling laws have made prediction routine for loss; the challenge is extending this success to capabilities, biases, robustness, and safety-relevant behaviors. We articulate requirements for such theories grounded in the history and philosophy of science, examine progress in mechanistic interpretability, fairness, memorization, and simplicity bias, and identify concrete open problems.

## 1. Introduction

Consider a tennis ball thrown into the air. Knowing the ball's position, velocity, and acceleration is sufficient to predict its motion with stunning accuracy. Add in higher order derivatives and a relatively simple differential equation will give the position of the ball until the end of the universe, with the primary source of error being the level of precision with which we can record the state of the rest of the stuff in the universe at the moment that the ball is thrown. This predictive power was hard-won, grounded in developing an understanding of how forces like gravity and properties like inertia operate on objects. The equations of motion are not merely curves fit to past observations, they are consequences of a deeper theory about how the universe works on a fundamental level that gives rise to the physical phenomenon we see in the world around us every day.

What would it mean to have such a theory for AI? **In this position paper, we argue that the field is far from having a scientific theory of AI and must move beyond post-hoc fixes to study the training dynamics of models in order to develop one.** When models exhibit undesirable behaviors, the dominant response is to "fix it in post"—to apply post-training interventions like RLHF or input and output filtering—rather than seek to understand why the behavior emerged. By studying models as static artifacts, we miss the opportunity to come up with explanations about how aspects of training affect the final model. Understand how training works as a dynamic process will allow us to predict and design effective training procedures.

### 1.1. The "Fix It in Post" Mentality

In film production, the phrase "fix it in post" describes a common cost-cutting strategy: deferring problems during shooting and correcting them in post-production. In AI, an analogous mentality has taken hold. Large models are trained on massive datasets with limited curation and, when they don't behave as desired, people ask "how can I further finetune the model to eliminate this undesirable behavior?" rather than "why did the model develop this behavior and how could it have been avoided?"

The limitations of this approach are almost immediately apparent. There is extensive work detailing how safety finetuning can be shown to be highly brittle, such as by finetuning on benign data (Qi et al., 2024; Lo et al., 2024), asking questions in the past tense (Andriushchenko & Flammarion, 2025), or asking them in a low resource language (Yong et al., 2023). Not only is the state-of-the-art training approach for any given task to do finetuning, but it is

---

[1]EleutherAI [2]Max Planck Institute for Software Systems [3]Carnegie Mellon University [4]University of Oxford, Martian [5]Boston University [6]Harvard University. Correspondence to: Stella Biderman <stella@eleuther.ai>.

*Proceedings of the 43$^{rd}$ International Conference on Machine Learning*, Seoul, South Korea. PMLR 306, 2026. Copyright 2026 by the author(s).

vanishingly rare that any other effective methodologies exist. Despite the extensive effort to combat frequency biases in models, the underlying representational biases persist (Mickel et al., 2025). This highlights the fundamental limitations of approaching AI behavior as something to be patched rather than something to be understood.

---

**Example: Band-Aids for Racism**

When DALL-E 2 generated mostly white men for prompts like "CEO," OpenAI's fix was to secretly appending words like "Black" or "female" to user prompts (Baio, 2022; Offert & Phan, 2022). This produces superficially diverse outputs while leaving the underlying bias intact. This is highly reminiscent of issues with Google Photos, which still cannot recognize gorillas over a decade after Black users were infamously misclassified as apes (Grant & Hill, 2023). After years of unsuccessful remediation, Google ultimately disabled primate recognition altogether rather than figure out how to fix the root cause (Hern, 2018).

---

This pattern parallels the history of other scientific disciplines, where intervention has preceded understanding and failures were met with more aggressive tuning rather than deeper inquiry. Pre-germ-theory medicine optimized treatments without causal models of disease (De Kruif, 1926); behaviorist psychology shaped behavior while refusing to theorize about internal representations (Chomsky, 1959); early aviation compensated for instability with brute force rather than aerodynamics (Vincenti, 1990). In each case, empirical success masked a lack of explanatory mechanism.

History suggests that such phases do not end simply through refinement of the interventions themselves, but when latent mechanisms are articulated and the space of possible behaviors becomes constrained by theory. Early flight experiments largely relied on trial-and-error, with designers making incremental adjustments to wing shapes and control surfaces without a principled understanding of lift or stability (Vincenti, 1990). The Wright brothers advanced the field by systematically measuring aerodynamic forces in wind tunnel experiments, producing detailed empirical tables of lift and drag (Wright, 1901). However, a full theoretical account lagged behind. Only with the development of modern aerodynamics, based on fluid dynamics and informed by Isaac Newton's laws of motion, did flight become a predictable engineering discipline rather than an experimental art (Prandtl, 1923; Anderson, 2011). What changed was not simply optimizing to the desired outcome, but investing in a fundamental understanding of the underlying mechanisms that constrained the design space.

We argue that the current dominant paradigm in machine learning research seeks primarily to optimize models to perform well at specific benchmarks without contributing to a deeper understanding model training, and that the dominance of this attitude is a major blocker to developing scientific theories of AI.

## 2. What would a "Science of AI" Look Like?

One natural approach to envisioning a science of AI is to look to what is considered good practice in fields of science and see what can be imported into our field. In *the Essential Tension* (Kuhn, 1977), the philosopher of science Thomas Kuhn laid out six characteristics of good scientific practice.[1]

**Empirical Accuracy**    A scientific theory should be able to make correct empirical predictions about the world.

**Internal Consistency**    A scientific theory's claims and assumptions should not contradict one another.

**External Consistency**    Scientific theories should be consistent with other existing scientific theories.

**Scope**    A scientific theory's consequences should extend beyond the particular observations, laws, or phenomena it was originally designed to explain.

**Simplicity**    A scientific theory should rely on as few independent assumptions as possible. Note that this is *conceptual simplicity*, in the sense of Aristotle (c. 330 BCE) and Aquinas (1912), among others and not syntactic simplicity.

**Fruitfulness**    A scientific theory should disclose new phenomena or previously unrecognized relationships among known ones, opening productive lines of research beyond those it was built to address.

Immediately we see significant divergence from typical research in AI, so much so that it doesn't make sense to try to judge work by AI researchers along these dimensions. Most research in AI doesn't produce theories that are intended to provide predictive accuracy across diverse experimental contexts or reveal underlying truths about the way the world works. To build a science of AI, then, we must develop our practices so that it makes sense to judge AI research by the standards that one judges fields of science. In the rest of this section we describe conceptual approaches to doing AI research that we believe are key to setting the field on a more scientifically grounded course.

---

[1]In the original Kuhn identifies five characteristics, combining external and internal consistency. We follow subsequent work and distinguish between them.

## 2.1. Prediction, Not Just Description

Arguably the most important innovation in the history of science was the shift to center the ability to make predictions about *what will happen* should an experiment be run, and then providing the framework for verifying the prediction. Sometimes this takes the form of precise equations (e.g., the laws of motion in physics) and sometimes this takes the form of less exact heuristics and methodologies that nevertheless can be used to draw reliable inferences (e.g., the theory of evolution by natural selection in biology). These predictions have significant practical utility, but they're also the primary way by which we can judge the empirical accuracy of a theory or demonstrate it has a broad scope. When a theory makes correct empirical predictions about data we haven't seen yet, we can have stronger confidence in the theory's empirical accuracy. When a theory tells us what will happen in a context we haven't previously considered, we have an opportunity to judge how broad the scope is and (potentially) check its external consistency with the predictions of theories that were developed in that context. In the study of AI, we rarely make such predictions. We describe, in great detail, how a model behaves *after* training. But *before* training, we rarely ever make claims about what we expect a model to do.

> ### Example: Darwin's Moth
>
> In 1862, Darwin encountered the orchid *Angraecum sesquipedale*, with a nectar spur nearly 30cm long. He reasoned that if the orchid reproduces something must pollinate it, and that only an insect with a proboscis long enough to reach the nectar would do. He predicted such a moth must exist in Madagascar. Over four decades later, the moth was discovered, confirming the prediction (Arditti et al., 2012).

One notable exception is predicting pretraining loss with scaling laws (Hestness et al., 2017; Rosenfeld et al., 2020; Kaplan et al., 2020; Hoffmann et al., 2022). Given model size and a compute budget, final loss can be predicted with reasonable accuracy. Labs use this to plan training runs and to flag issues early in a training run if it isn't in line with predictions, which offers practical utility.[2] Recent work has also shown success at extending prediction to some evaluation benchmarks (Held et al., 2026; Chen et al., 2025; Grattafiori et al., 2024; Achiam et al., 2023). We can conceptually extend this paradigm to more generally ask, for a particular property $P$, what methodologies would enable us to predict the value of $P$ for a model given smaller models. While some research in this direction exists for memorization (Biderman et al., 2023a; Prashanth et al., 2025) and

___
[2]We are not aware of any clear public writings on this topic, and this comment is based on personal communication.

social bias (Biderman et al., 2023b; Patel et al., 2024), we view the ability to develop these theories as an essential component of the science of AI.

## 2.2. Models as Time-Evolving Processes

Scientific theories provide answers to "why" questions. While fields such as interpretability often claim to answer such questions for AI, typical work misses something crucial. Answering the question "why did a model do X on Y input" certainly has some utility (e.g. for corporations interested in product assurance, user-satisfaction, and compliance), but on a scientific level it's fundamentally limited. A more scientific mindset would be to ask "why did the model develop this behavior?" This involves shifting from viewing models as static objects to viewing them as snapshots of time-evolving processes and studying the entire dynamical system (Saphra, 2023; Biderman et al., 2023b; Hoogland et al., 2023). When the object of study is the training process rather than the finished model, an account of that process can be applied any model it produces, not to one specific set of weights (Sellam et al., 2022).

One approach examines training step-by-step. Olsson et al. (2022) discovered that induction heads form abruptly during a phase transition early in training. Saphra & Lopez (2019) showed that representations develop in a characteristic order, with syntax emerging before semantics and Chen et al. (2024a) showed that internal syntactic structure precipitates the acquisition of grammatical rules. Kangaslahti et al. (2026) found that smooth aggregate loss curves hide discrete skill-specific breakthroughs, visible only when loss is decomposed by sample type.

A complementary approach varies training conditions across runs. Qin et al. (2024) train models on data with varying syntactic complexity, finding that data composition determines whether models learn hierarchical rules or surface shortcuts. A number of results have likewise found that models can only systematically compose concepts which appear in diverse contexts during training (Allen-Zhu & Li, 2025; Okawa et al., 2023; Chang et al., 2025).

Understanding these training trajectories should matter, even to people solely interested in the final model performance, because it enables *intervention during training*. If a model is trending toward memorizing sensitive content, developers could adjust the data before memorization solidifies. If an undesirable bias is emerging, the data mixture could be re-balanced (Biderman et al., 2023b). Biderman et al. (2023a) showed that intermediate checkpoints predict final memorization better than smaller fully-trained models, suggesting that the trajectory contains information that endpoint comparisons miss.

## 2.3. Properly Identifying Objects of Study

Every scientific theory must choose what to treat as a phenomenon to be explained, what to consider a confounder, and what to treat as primitive. Is temperature a parameter to a thermodynamic system, or is it a measure of particle movement? Do we inherit eye color, or do we inherit genes which control the proteins that determine eye color? As science advances, the primitives of one theory become the results of a deeper one.

In AI we face analogous choices, but these choices are often made implicitly rather than deliberately (Ayonrinde & Jaburi, 2025). When we study a phenomenon across models with different random seeds, we are implicitly declaring that seed-specific details are not part of our explanation: they are noise to be averaged over, not signal to be accounted for.

This is not merely a theoretical concern. Even high-level findings can be sensitive to random seeds (Sellam et al., 2022), and behaviors identified in one model may not appear in another trained on slightly different data (Qin et al., 2024). Near "emergence" thresholds, random variation leads to clusters of generalization behavior (Zhao et al., 2024) which each correspond to different loss basins (Juneja et al., 2023) and internal mechanisms (Li et al., 2025). Sparse autoencoders trained with different random initializations can learn different features (Paulo & Belrose, 2025), revealing limitations for using them as a proxy for a full model. Without systematic studies of variation, we cannot know whether we are describing the results of fundamental phenomena or implementation details.

---

**Example: Data Attribution**

Given a trained model, its training data, and a behavior of interest, the goal of data attribution is to identify the training examples most responsible for that behavior. Data encountered later in training has a larger influence on model behavior, so the answer depends on what is held fixed: a method that conceptualizes the task as reshuffling the data and retraining (Ilyas et al., 2022; Park et al., 2023) and one that holds the realized run fixed and asks what that specific model owes to each example (Ilyas & Engstrom, 2025) are answering different questions. Unfortunately, its common for work to confuse or conflate these settings (Deng et al., 2024; Mlodozeniec et al., 2025; Wang et al., 2026).

---

This logic extends beyond random seeds to other dimensions of variation. Most NLP research studies English-centric models, yet claims to study "language models" rather than "English language models". This common oversight led to the "Bender Rule" (Bender, 2019): that research in natural language processing should explicitly state which languages

are studied.

For models where training data are released or described, training data often consists of at least 90% English[3] data (Brown et al., 2020; Grattafiori et al., 2024), but claims about them are usually about learning language generally. At the same time, research on non-English languages is seen to be uninformative to field more broadly. In fact, recent results suggest that models trained in exactly the same way learn some languages more easily than others (Cotterell et al., 2018; Arnett & Bergen, 2025), and model design decisions impact different languages differently (Gerz et al., 2018; Arnett et al., 2025; Shani et al., 2026). A deliberate science of AI would make explicit choices about which dimensions of variation matter for which questions, and then systematically test those choices.

## 2.4. Predict, Intervene, Design: Progress towards our Goals for a Science of AI

How can we tell if we are making progress towards developing a science of AI? We cannot measure a theory the way we can measure a benchmark score, so we need some concrete way to tell whether we are acquiring one.

A useful move is to ask what such an understanding should empower us do, and to treat our growing ability to do it as a measure of progress. We identify three capabilities, each more demanding than the last and each building on the one before. The first is *predicting training*: forecasting a property of a model from its training setup, before the run finishes and ideally before it begins, requiring that we can make reliable predictions (Section 2.1). The second is *intervening on training*: recognizing that a trajectory is heading somewhere undesirable and redirecting it while training is still underway, requiring an understanding of models that views them as a time-evolving dynamical process (Section 2.2). The third is *designing training*: fixing the properties we want in advance and constructing a procedure that reliably produces them, requiring that we have correctly identified which behaviors are stable enough to target (Section 2.3). Each is harder than the last because each requires everything the one before it did, and then more.

These capabilities operationalize progress, but its important to remember that they do not define the goal. The goal is a scientific theory that explains why training produces the behaviors it does. The capabilities themselves are not the goal but instead are how we would recognize that we are making progress towards a predictive theory. We could predict a property by fitting a curve to past runs without understanding what drives it, and we could in principle design

---

[3]Increasingly, powerful models are trained on high proportions of English and Chinese data, e.g. DeepSeek (DeepSeek-AI et al., 2025), but the same issue remains as claims are rarely limited to English-Chinese language models.

a procedure that yields a desired property by searching over enough configurations, with no account of why it works. Abilities like these are necessary signs of understanding, since a theory that predicts nothing and guides no intervention is hardly a theory. They are not sufficient for it. A predictor or a recipe that explains nothing is precisely the kind of opaque success this paper argues against.

Treating the capabilities as progress rather than as the destination also clarifies why the program is worth pursuing before any complete theory arrives. Each is valuable on its own as we acquire it: a reliable prediction saves a failed run, a timely intervention keeps a harmful behavior from setting in, and even partial design makes outcomes more controllable. These are the returns collected along the way, and they are why a science of training dynamics is not a luxury to be deferred until the theory is finished.

The case studies in Section 3 can be read as a progress report of this kind, measuring different areas against the three capabilities. Mechanistic interpretability includes some of the most promising work here, studying how circuits form across training and confirming their role through intervention. But its mainstream is heavily influenced by the production needs of companies and by anxieties about advanced AI toward compliance-style accounts of why a finished model produced a given output. Fairness is further behind and seems largely uninterested in catching up. Memorization is another field that has seen a lot of success building towards a scientific approach lately, with work focused on making and validating predictions and on developing theory-driven understanding of why memorization happens. Work on simplicity bias is assembling the mechanistic pieces an explanation would need. Measured this way, progress is real but uneven, concentrated in a few areas and barely begun in others.

## 3. Case Studies

We now examine several research areas through our proposed lens, identifying promising directions and significant gaps in contemporary research practice.

### 3.1. Mechanistic Interpretability

Mechanistic interpretability aims to reverse-engineer neural network computations, identifying circuits responsible for specific behaviors. This goal aligns naturally with scientific aspirations: identifying mechanisms is precisely the causal, explanatory work that distinguishes science from description. However, most mechanistic interpretability work remains fundamentally descriptive. It answers "what computations does this model perform?" rather than "why did these computations develop?" or "how can we instill specific mechanisms?"

The first step towards answering these questions is to find generalizable patterns. Most circuit analysis studies a single model—often a single checkpoint. When circuits are identified in GPT-2 Small, we typically do not know whether they exist in GPT-2 Medium, in a different random seed, or at intermediate training steps. Tigges et al. (2024) directly address this gap, studying circuit consistency across checkpoints and scales. Their finding is nuanced: high-level mechanisms are consistent, but which specific neurons participate fluctuates considerably. Similarly, Rivière & Trott (2025) find that the layers of the attention heads most responsible for sense disambiguation performance varies across random seeds. If individual neuron participation is unstable, then "neuron 347 in layer 8 detects indirect objects" is the wrong level of description—it captures an implementation detail rather than a stable phenomenon. A scientific theory must characterize mechanisms at abstraction levels that are invariant. And in a variety of settings, even the high-level mechanisms can vary widely, leading to variation in their resulting edge case behavior (Li et al., 2025; Juneja et al., 2023; Huang et al., 2025).

By studying mechanisms across multiple settings (per Section 2.3), researchers can confirm hypotheses about their connection to specific model behavior. Chen et al. (2024a), noting that language models developed internal syntactic structure suddenly, manipulated the timing of this onset to confirm that it precipitated a subsequent breakthrough in grammatical capabilities. Li et al. (2025) trained hundreds of small models, correlating specific attention patterns with intuitive out-of-distribution behaviors. All of these approaches confirmed a link between the mechanism and its associated behavior.

One success story for training analysis (as proposed in Section 2.2) is arguably the most well-studied mechanism in LLMs: induction heads. Olsson et al. (2022) first showed that induction heads formed alongside an increase of in-context learning capabilities. Subsequent results have deepened and complicated this link further; in-context learning can be limited to an early stage of training (Singh et al., 2023) and as LLMs memorize each task in their weights, they rely less on induction heads to execute it (Yin & Steinhardt, 2025). When an association is thus confirmed throughout training and across different settings, we can be confident in the link.

But most work in mechanistic interpretability neglects the ultimate cause of observed structures. Without such confirmation, interpretability claims can be misleading. For example, selective neurons (i.e. neurons that activate for specific output classes; Zhou et al., 2015) were a promising early candidate for interpretability through monosemanticity. Unfortunately, these monosemantic neurons were revealed to damage model performance (Leavitt & Morcos, 2021), a

result explained when Ranadive et al. (2023) showed them to be a vestigial remnant of early training. Rather than providing a viable interpretability method in performant models, selectivity revealed a side effect of the training process.

> **Open Problem: Differentiating Between Computation and Data Structure**
>
> One simple way of using the training process in interpretability is to compare a trained model with its random initialization. Both SAEs (Heap et al., 2025) and saliency maps (Adebayo et al., 2018) provide plausible interpretations at initialization, calling these methods into question as ways of understanding a trained model. These results also reveal a general problem when analyzing structure in hidden representations: which structures are surface-level patterns in the input data, which actually illuminate the model's processing of those inputs, and which are best ascribed to minds of the people doing the research (Bolukbasi et al., 2021; Méloux et al., 2025; Ayonrinde & Jaburi, 2025)?

Interpretability methods can also create an illusion of understanding when explanations are only validated on the same distribution that produced them. Bolukbasi et al. (2021) show that individual neurons and linear directions in BERT can appear to encode simple semantic concepts, even when those apparent concepts reflect the geometry of the representation space and the narrow support of the datasets being inspected. In this sense, an interpretation can be locally compelling while failing to identify a stable mechanism. A related problem appears for feature-attribution explanations: Dombrowski et al. (2019) show that explanations can be arbitrarily manipulated by small input perturbations that leave the model's output nearly unchanged, tying this instability to the geometry of neural-network decision surfaces. Both results suggest that explanations should not be trusted merely because they are sparse, semantically appealing, or visually plausible.

Out-of-domain generalization provides a natural stress test for such claims. If an interpretation identifies a real mechanism rather than a dataset artifact, it should predict behavior not only on the examples used to discover it, but also under controlled distribution shifts where the hypothesized mechanism is preserved and competing shortcuts are broken. This point connects mechanistic interpretability to the broader literature on shortcut learning: models often achieve high in-domain accuracy by relying on features that are predictive in the benchmark but non-causal for the task (Geirhos et al., 2020). McCoy et al. (2019) show that NLI models trained on standard datasets can rely on syntactic heuristics that fail during controlled evaluation, despite strong in-domain performance. Similar concerns apply to explana-

tions themselves: Chrysostomou & Aletras (2022) find that common faithfulness metrics for post-hoc explanations can behave misleadingly in out-of-domain settings, motivating explicit random baselines and distributional tests. Thus, interpretability claims should be evaluated as scientific hypotheses: they should generate predictions about when a model will generalize, when it will fail, and which interventions would change that behavior.

### 3.2. Fairness and Bias

An extensive literature documents how biases in training data propagate into learned models. Word embeddings exhibit gender stereotypes in occupational associations (Bolukbasi et al., 2016); facial recognition systems show dramatically different error rates across demographic groups (Buolamwini & Gebru, 2018); recidivism algorithms have higher false positive rates for minoritized racial groups (Angwin et al., 2016). This work has been valuable for awareness and evaluation metrics, yet it remains largely diagnostic. Practitioners measure bias after training and attempt mitigation through resampling, reweighting, or constrained optimization, but cannot answer the forward-prediction question of Section 2.1: given a target distribution, what training data modifications would produce a model exhibiting it?

Without predictive theories mapping data characteristics to model behavior, fairness interventions remain empirical trial-and-error. Practitioners cannot reason forward from dataset design to downstream consequences, heavily limiting the potential impact of this work. Sellam et al. (2022) and Biderman et al. (2023b) demonstrate how their model suites can enable more rigorous scientific study of these phenomena, but the gender bias literature is yet to build on them.

> **Open Problem: Designing Social Bias**
>
> Suppose one decides *a priori* that an image generation model should generate a woman when prompted to generate a person of a given profession with frequency equal to the U.S. Bureau of Labor Statistics' measured data. What should the rate of women in the training data be to accomplish this? Setting it equal to the target rate is insufficient (Zhao et al., 2017; Seshadri et al., 2024; Chen et al., 2024b; Roos et al., 2026), but there is currently no training method that results in the desired behavior.

We know models exhibit biases, and we know training data contains imbalances, but we lack studies that quantitatively connect the two across the full pipeline. Hall et al. (2022) show that bias amplification occurs primarily when recognizing group membership is easier than recognizing class membership, and that amplification correlates with model

capacity and overconfidence—but this work focuses on discriminative classifiers in controlled settings. For LLMs, each training stage could amplify, preserve, or dampen input bias through different mechanisms: pretraining may bake in statistical associations (Bender et al., 2021), while RLHF can either mitigate bias through preference learning or introduce new biases from annotator demographics (Kirk et al., 2024). Recent work suggests that bias introduced during pretraining persists through fine-tuning (Ghate et al., 2025), indicating that *post hoc* interventions may be fundamentally limited. Furthermore, as synthetic data becomes a larger fraction of training corpora (Alemohammad et al., 2024), we need to understand whether synthetic generation launders bias (making it harder to detect) or compounds it through feedback loops (Mehrabi et al., 2024). Without causal tracing through the pipeline, fairness interventions remain guesswork.

---

**Open Problem: Cultural and Linguistic Erasure**

Model behavior varies significantly across languages, dialects, cultures, and even names associated with different backgrounds, resulting in uneven product quality and downstream harms (Blodgett et al., 2020). Generative models can flatten or distort marginalized cultures, and hallucinations can create misinformation about communities, histories, or practices (Ortu et al., 2026). Yet we lack systematic measurement infrastructure: which cultures are most prone to erasure or distortion? Can we predict, before deployment, which cultures and communities will experience representation harms, and how these harms vary with model scale, training data composition, and post-training choices?

---

Global MMLU is a step toward this kind of measurement infrastructure: it finds that 28% of questions require culturally sensitive knowledge, and shows that models optimized for translated benchmarks risk overfitting to Western-centric concepts (Singh et al., 2025). But evaluation alone does not explain where these failures enter the pipeline. Addressing cultural and linguistic erasure therefore requires both better benchmarks that test cultural knowledge across many communities and causal studies connecting training choices to downstream cultural appropriateness.

### 3.3. Memorization

Memorization research offers a compelling example of nascent theory-building, with recent work establishing both predictive frameworks and causal methodologies that exemplify the scientific approach we advocate. The memorization case study in Pythia (Biderman et al., 2023b) explicitly framed their work as scientific prediction: they hypothe-

sized, based on intuitions about training dynamics, that data encountered later in training would be more likely memorized. Initial experiments suggested order-independence, but subsequent work with refined methodology confirmed the original hypothesis (Lesci et al., 2024; Kuditipudi et al., 2026). This progression exemplifies the scientific process we advocate: (1) hypothesis stated *before* experiments, (2) grounded in a causal story about training dynamics, (3) initial null result led to refined methods rather than abandonment, (4) finding generalizes across models. This is an instantiation of treating models as time-evolving processes (Section 2.2): a claim about the training trajectory is stated up front and tested across models and through different versions.

A second line of work has established rigorous methodology for studying *how much* repetition is required for verbatim memorization. Huang et al. (2024) developed a causal intervention framework to study verbatim memorization in controlled settings. Using cross-model interchange interventions, they show that non-trivial repetition is necessary for true memorization and that it cannot be attributed to specific weights alone, but is fundamentally tied to general language modeling capabilities. Apparent cases of "single-shot" memorization are often reconstructions of templated texts (Prashanth et al., 2025) or frequent patterns rather than genuine memorization. Additionally, over half of the interventions producing memorized tokens rely only on general language modeling components, explaining why unlearning attempts often degrade overall model quality. These interchange interventions transfer across models, exemplifying the generalization recommendations of Section 2.3.

Membership inference studies provide converging evidence. Duan et al. (2024) found that attacks across Pythia models barely outperform random guessing, due to massive datasets, near-one-epoch training, and fuzzy member/non-member boundaries. This suggests that, at scale, most sequences are not memorized in ways that leave detectable traces, consistent with findings that substantial repetition is required for verbatim memorization which is further supported by findings from the Hubble models (Wei et al., 2026).

Despite the significant effort put towards developing causal analyses of memorization, significant and very basic gaps remain in our understanding of memorization dynamics. For example:

**Open Problem: Predicting Memorization**

Biderman et al. (2023a) introduce a new question about memorization: given a dataset in a fixed order, can you predict *which specific sequences* will be memorized by a large model trained on that data without training the full model? Biderman et al. (2023a) analyze using both smaller and partially trained models to forecast memorization, concluding that their method was insufficient and challenge future work to do better. While this work has been replicated (Liu et al., 2024) and influenced subsequent work on related questions (Dentan et al., 2024; Lesci et al., 2024; Prashanth et al., 2025), the core question is still open.

Memorization is not simply a function of repetition and recency. Recent work reveals a surprising non-monotonicity: models are most "absorbent" around 10–20% into training, despite having more unused capacity earlier (Duan et al., 2024). Related evidence appears in Masud et al. (2024), who find that finetuning early RoBERTa checkpoints works best for hate speech classification. This suggests an interplay between three factors: (1) remaining capacity, (2) linguistic competence (early models may lack the representational structure to encode verbatim sequences), and (3) data distribution (what else competes for the same capacity at each training stage). Additionally, Liu et al. (2024) show that memorization scores spike at the end of training and that earlier data is progressively forgotten unless revisited, while later checkpoints memorize more readily even for out-of-distribution sequences (Duan et al., 2024). A unified theory predicting memorization rates from these factors jointly would enable principled decisions about when to introduce sensitive data during training, and whether capacity-limited models can be "safely" trained on data that larger models would memorize.

### 3.4. Simplicity Bias and Learning Dynamics

Research on distributional simplicity bias represents a promising foundation for scientific theory. Refinetti et al. (2023) established that networks trained with SGD learn statistical features of increasing complexity: first differences in class means, then covariance structure, only later higher-order cumulants like co-skewness. Belrose et al. (2024) extended this to real data, demonstrating characteristic U-shaped loss curves confirming progressive sensitivity to higher-order statistics. More recently, Chang & Bergen (2025) provided mechanistic evidence that language models learn a bigram subnetwork early in training. This is consistent with findings in Michaelov et al. (2025), which show that language model predictions correlate with unigram probabilities very early in training. Shortly after, they

correlate with bigrams, then trigrams, and so on. These patterns hold across model scales and architectures.

A related line studies spectral bias: the tendency to learn low-frequency functions before high-frequency ones (Rahaman et al., 2019; Xu et al., 2020; Ronen et al., 2019). Both concern the order of feature learning, but are not straightforward generalizations of each other. Spectral bias concerns Fourier decomposition of the learned function; distributional simplicity bias concerns statistical moments of data distributions. It is natural to ask whether there is a potential unification of these theories. For example, it is possible that the real bias is simply towards patterns that occur in training data disproportionately often and *nature* (or our data collection methods) have a bias towards patterns easily explained in terms of lower order statistics and low-frequency signals.

**Open Problem: simplicity bias → OpenFold**

The OpenFold paper (Ahdritz et al., 2024) trains models to predict protein stuctures and finds something remarkable: the model seems to learn spatial dimensions sequentially over the course of training and the predictions produced by checkpoints saved early in training are approximately PCA optimal projections of the final prediction into lower dimensions. This seems quite evocative of research on simplicity biases, but its unclear which simplicity bias literature it connects to. Are the empirical results reported in Ahdritz et al. (2024) consistent with theoretical literature, and if so could these results have been predicted in advance?

## 4. Alternative Views

Our argument that AI research should prioritize predictive, causal theories of training dynamics runs against several influential perspectives in the field. In this section, we outline these alternative viewpoints and explain where we agree, where we disagree, and what is ultimately at stake.

**Engineering Sufficiency.** A common view holds that AI does not require deeper scientific theory: large-scale experimentation, benchmarking, and post-hoc fixes are sufficient as long as systems improve empirically. This perspective is supported by the undeniable success of current engineering practice (Krizhevsky et al., 2012; He et al., 2016; Brown et al., 2020; Chowdhery et al., 2023). Our claim is not that such methods fail, but that they optimize locally and opaquely. Without predictive theories, failures cannot be anticipated, interventions do not reliably generalize, and success at one scale offers limited guidance at the next (Liu et al., 2025; Lourie et al., 2025). Engineering has made real but fragile advances, and by supplementing it with scientific

rigor we can cement the gains from engineering-focused work in a more reliable and robust fashion.

**Anti-Theory Skepticism.** Some argue that neural networks are too complex for meaningful theory, or that the stochastic nature of training makes it impossible (Baldassi et al., 2016; Pontin, 2018; Adolfi et al., 2024). We agree that AI theories might be approximate, heuristic, and domain-specific. Still, empirical regularities such as scaling laws, memorization dynamics, and simplicity bias show that even partial theories can yield predictive insight (Kaplan et al., 2020; Chen et al., 2024a; Biderman et al., 2023a;b; Chen et al., 2025; Tao et al., 2024; Held et al., 2026). The key question is whether such theories constrain expectations and guide interventions better than post-hoc description, even if they remain imperfect. Furthermore, other fields such as thermodynamics, quantum mechanics, and genetics have shown that even highly non-deterministic processes can be fruitfully analyzed through statistical tools.

**Automation as Scientific Progress.** A growing view holds that the path to AI understanding is to automate research itself: using AI systems to generate hypotheses, run experiments, and discover patterns that may be opaque to humans (Lu et al., 2026; Yamada et al., 2025). On this account, scientific insight need not be human-interpretable as long as automated processes reliably produce better models. While such tools can accelerate discovery, this stance risks conflating optimization with understanding. If the research process itself becomes a black box, we gain neither predictive theories nor principled control. Instead, there is faster iteration on artifacts whose failure modes remain poorly understood.

**Safety Pragmatism.** In safety and alignment work, an influential view prioritizes immediate harm reduction through post-training interventions (Ouyang et al., 2022; Bai et al., 2022), arguing that waiting for deeper understanding is impractical given deployment pressures. This concern is legitimate, and short-term mitigations are often necessary. However, many recurring safety failures (e.g. jailbreaks, regressions, brittleness under scale) are a direct result of the "fix it in post" attitude (Wei et al., 2023; Zou et al., 2023) and offer no path toward design-level solutions that would prevent problems from arising. A science of training dynamics would not replace pragmatic safeguards but supplement them with more robust methods as they are developed, such as O'Brien et al. (2026) and Cloud et al. (2024).

## 5. Conclusion

We have argued that AI research should aspire to genuine scientific theories: causal explanations that predict novel phenomena, generalize across instances, and unify disparate observations. The field's current emphasis on *post-hoc* analysis and post-training fixes—the "fix it in post" mentality—while sometimes practically necessary, cannot substitute for fundamental understanding.

The building blocks exist. Scaling laws demonstrate that prediction is achievable: we can forecast loss from early signals and design compute allocations accordingly. Work on simplicity bias shows causal mechanisms can be identified. Released checkpoints and datasets enable the longitudinal studies that dynamical understanding requires. What is needed is extending scaling laws' success to the properties we care about. We envision a theory that allows us to first *predict* outcomes from early training signals, then *intervene* when trajectories go wrong, ultimately *design* training procedures that guarantee desired properties. Scaling laws have achieved this for loss; the challenge is achieving it for capabilities, biases, and safety.

The problems we have outlined are difficult—but they are the problems that a mature science of AI will solve.

## Acknowledgments

This work was informed by conversations with Jennifer Mickel, Aviya Skowron, Isabelle Lee, and Louis Jaburi.

N.S. is supported by a Technical AI Safety Research Grant from Coefficient Giving via Berkeley Existential Risk Initiative. This work has been made possible in part by a gift from the Chan Zuckerberg Initiative Foundation to establish the Kempner Institute for the Study of Natural and Artificial Intelligence at Harvard University.

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
