# OpenReview forum: "Position: Don't Just "Fix it in Post'': A Science of AI Must Study Learning Dynamics"
_ICML.cc/2026/Position_Paper_Track — ICML 2026 Position Paper Track spotlight_

### Official Review · Reviewer_JXAd · 2026-02-27

**Significance:** 4
**Argument Clarity:** 3
**Rating:** 5
**Confidence:** 4

**Questions:**

**Questions**
1. pg.4 column 1 "features prevalent in training data are learned first". As opposed to features not present in the training data being learned, I'm not sure I understand this point?

**Alternative Views Section:**

Yes

**Compliance With Llm Reviewing Policy A Conservative:**

Affirmed.

**Discussion Potential:**

4

**Final Justification:**

The main weakness I identified has been addressed to my satisfaction. With the alternative views section improved I recommend accepting this paper.

**Paper Summary:**

The authors argue that current AI research is overly post-hoc: RLHF to fix alignment issues, addition of terms to prompts to try to curtail unwanted behavior, filtration of outputs. Instead they argue that the field should adopt a "predict, intervene, design" framework, where we aim to predict downstream results from early on in training or smaller experimental runs, can intervene during training if something is going wrong, and can specify properties of the model we want before we even start training.

**Position:**

Yes

**Position In Title:**

Yes

**Related Work:**

2

**Strengths And Weaknesses:**

**Strengths**
- I felt that this was a very clearly argued position paper, thanks to the examples and "success stories" it was very easy for me to understand the argument for the stated position.
- I think this paper is likely to generate some discussion about the insufficiency of current alignment/safety techniques.
- The case studies section does a good job of making a case for the importance of the concerns raised and the research agenda section is very constructive which I liked.





**Weaknesses**
- The alternative views section is a bit sparse, not including any citations. It would be nice to see some examples of these contrary views. This is quite a large weakness of the paper for me and were it addressed I would recommend accepting this paper.

**Minor Weaknesses**
- On pg. 3 2nd column "If we detect mid-training that a model is trending toward memorizing sensitive content,"
-On pg.4 1st column 'Most AI research studies English-centric models, yet claims to study language models” rather than English models.” '

**Support:**

3

---

> ### Author Rebuttal · Authors · 2026-03-31
>
> We thank the reviewer for their feedback on our paper. We are glad the reviewer felt the examples and success stories were helpful.
>
> We also agree with the reviewer that the Alternative Viewpoints section would benefit from stronger support, as provided below:
>
> Our argument---that AI research should prioritize predictive, causal theories of training dynamics---runs against several influential perspectives in the field. In this section, we outline these alternative viewpoints and explain where we agree, where we disagree, and what is ultimately at stake.
>
> **Engineering Sufficiency.** A common view holds that AI does not require deeper scientific theory: large-scale experimentation, benchmarking, and *post hoc* fixes are sufficient as long as systems improve empirically. This perspective is supported by the undeniable success of current engineering practice. Our claim is not that such methods fail, but that they optimize locally and opaquely. Without predictive theories, failures cannot be anticipated, interventions do not reliably generalize, and success at one scale offers limited guidance at the next [1,2]. Engineering has made real but fragile advances, and by supplementing it with scientific rigor we can cement the gains from engineering-focused work in a more reliable and robust fashion.
>
> **Anti-Theory Skepticism.** Some argue that neural networks are too complex for meaningful theory, or that the stochastic nature of training makes it impossible [3,4]. We agree that AI theories might be approximate, heuristic, and domain-specific. Still, empirical regularities such as scaling laws, memorization dynamics, and simplicity bias show that even partial theories can yield predictive insight [5,6]. The key question is whether such theories constrain expectations and guide interventions better than *post hoc* description, even if they remain imperfect. Furthermore other fields such as thermodynamics, quantum mechanics, and genetics have shown that even highly non-deterministic processes can be fruitfully analyzed through statistical tools.
>
> **Automation as Scientific Progress.** A growing view holds that the path to AI understanding is to automate research itself: using AI systems to generate hypotheses, run experiments, and discover patterns that may be opaque to humans [7,8]. On this account, scientific insight need not be human-interpretable as long as automated processes reliably produce better models. While such tools can accelerate discovery, this stance risks conflating optimization with understanding. If the research process itself becomes a black box, we gain neither predictive theories nor principled control. Instead, there is faster iteration on artifacts whose failure modes remain poorly understood.
>
> **Safety Pragmatism.** In safety and alignment work, an influential view prioritizes immediate harm reduction through post-training interventions, arguing that waiting for deeper understanding is impractical given deployment pressures [9,10]. This concern is legitimate, and short-term mitigations are often necessary. However, many recurring safety failures---jailbreaks, regressions, brittleness under scale---are failures of prediction [11,12]. *Post hoc* fixes attempt to intervene without the predictive understanding that would make intervention robust and reliable.  These fixes offer no path toward design-level solutions that would prevent problems from arising.
>
> [1] Xu, Wenrui, and Keshab K. Parhi. "A survey of attacks on large language models." (2025).
>
> [2] Lourie, Nicholas, Michael Y. Hu, and Kyunghyun Cho. "Scaling laws are unreliable for downstream tasks: A reality check, 2025."
>
> [3] Baldassi, Carlo, et al. "Unreasonable effectiveness of learning neural networks: From accessible states and robust ensembles to basic algorithmic schemes." PNAS (2016).
>
> [4] Pontin, Jason. "Greedy, brittle, opaque, and shallow: The downsides to deep learning." Wired.com (2018).
>
> [5] Kaplan, Jared, et al. "Scaling laws for neural language models." (2020).
>
> [6] Chen, Angelica, et al. "Sudden drops in the loss: Syntax acquisition, phase transitions, and simplicity bias in MLMs." ICLR (2023).
>
> [7] Lu, Chris, et al. "The ai scientist: Towards fully automated open-ended scientific discovery." (2024).
>
> [8] Yamada, Yutaro, et al. "The ai scientist-v2: Workshop-level automated scientific discovery via agentic tree search." (2025).
>
> [9] Ouyang, Long, et al. "Training language models to follow instructions with human feedback." NeurIPS (2022): 27730-27744.
>
> [10] Bai, Yuntao, et al. "Constitutional ai: Harmlessness from ai feedback." (2022).
>
> [11] Wei, Alexander, Nika Haghtalab, and Jacob Steinhardt. "Jailbroken: How does llm safety training fail?." NeurIPS (2023).
>
> [12] Zou, Andy, et al. "Universal and transferable adversarial attacks on aligned language models." (2023).

---

> > ### Author Rebuttal · Reviewer_JXAd · 2026-04-02
> >
> > Thank you for the considered rebuttal.
> >
> > My main concern was with the weakness of the alternative viewpoints section. As this has been addressed I will revise my score.

---

### Official Review · Reviewer_Zi6w · 2026-03-08

**Significance:** 3
**Argument Clarity:** 3
**Rating:** 5
**Confidence:** 4

**Questions:**

1. On the recommendation to test generalization systematically, isn't this already standard in the review process? Reviewers almost always request additional experiments or evaluations to support a paper's claims.

2. Neural scaling laws were largely discovered empirically by practitioners. Doesn't this support the view that progress in AI has so far been driven primarily by empirical exploration rather than theory? I'm aware that there are rigorous analyses (e.g., using random matrix theory) that explain certain scaling phenomena under restrictive assumptions, but that does not change the fact that the predictive utility of scaling laws was not originally established by theory.

**Alternative Views Section:**

Yes

**Compliance With Llm Reviewing Policy A Conservative:**

Affirmed.

**Discussion Potential:**

3

**Final Justification:**

My concerns are addressed so I keep my positive evaluation.

**Paper Summary:**

This paper argues that much of current AI research treats AI models (e.g., LLMs) as static objects and focuses on post hoc analyses of specific behaviors at a final checkpoint. In contrast, the paper advocates for a more scientific understanding of AI that focuses on learning dynamics and regards models as time-evolving systems shaped by data and training procedures. Then the authors propose a predict–intervene–design hierarchy and use a series of case studies (including topics such as fairness, memorization, simplicity bias and so on) to illustrate why focusing on learning dynamics is necessary. The paper also mentions some open challenges at each layer of the hierarchy and discusses several alternative views.

**Position:**

Yes

**Position In Title:**

Yes

**Related Work:**

3

**Strengths And Weaknesses:**

**Strengths.**

1. The paper supports its position with many concrete examples and well-chosen analogies from other scientific disciplines, which motivates why causal analysis of learning dynamics matters for understanding AI systems.

2. Some suggestions, particularly the methodological recommendations, are indeed helpful for shaping how the community studies models.

3. The discussion of alternative viewpoints is thoughtful and open-minded. The authors discuss opposing perspectives with an open tone, which would make the argument in this paper easier to accept for a broader audience.

**Weaknesses.**

1. I do believe the paper's position is important and worth emphasizing. That said, the core message, namely studying learning dynamics ( or training trajectories and loss landscape) rather than only post hoc behavior, already seems widely recognized in large parts of the theory communities and is reflected in many feature learning based works. In practice, fully characterizing learning dynamics for modern models is extremely hard, so post hoc methods are more like a temporary compromise. From this angle, the proposed predict–intervene–design hierarchy may feel less novel to readers already working on theoretical foundations, since many would naturally aim for more deeper understanding as soon as suitable mathematical tools become available.


2. This point is more of a suggestion than a weakness. The paper could strengthen its case by discussing negative results and hardness or impossibility perspectives. At present, it can be difficult to convince practitioners that theory will directly guide model or training pipline design, but they may be easier convinced by theory that proving what is impossible (thus ruling out unproductive directions). There is a rich tradition of such results, e.g., impossibility results in domain adaptation and fairness by Shai Ben-David and his collaborators, that could provide an additional, complementary justification for the predict–intervene–design agenda.

**Support:**

3

---

> ### Author Rebuttal · Authors · 2026-03-31
>
> We thank the reviewer for their thoughtful and supportive evaluation, and we appreciate the constructive suggestions. We are glad that the reviewer finds the position important, the examples well-motivated, and the alternative views fairly presented.
>
> **W1:**
>
> *On the perceived lack of novelty.*
> We agree that parts of the theory community already recognize the importance of studying learning dynamics. However, that belief is quite niche within machine learning more broadly and largely absent from AI interpretability research. In our experience, much of current practice still treats models as static artifacts and prioritizes post-hoc analysis. This gap between theoretical awareness and mainstream practice is precisely what motivates the paper.
>
> Our contribution is not simply advocating for studying learning dynamics in isolation, but articulating the unifying predict–intervene–design framework, which makes explicit how empirical, theoretical, and methodological strands of work can be connected in a cumulative scientific program. Even if individual components are familiar, we believe their synthesis and positioning are novel and valuable.
>
> *On feasibility and “waiting for better tools.”*
>
> We fully agree that characterizing learning dynamics in modern models is challenging. However we believe currently existing tools are significantly underutilized. Consider the example of memorization and training order. Biderman et al. (2023b) and Lesci et al. (2024) don’t obtain contradictory answers because new tooling was invented. All of the analysis in Lesci et al. (2024) could have been done in Biderman et al. (2023b), and they only analyze models and data released in Biderman et al. (2023a) and Biderman et al. (2023b). At a NeurIPS 2025 workshop talk, Stella Biderman publicly credited her incorrect conclusions to her insufficient statistical understanding. While some problems certainly require new tooling and fundamental ideas, we believe that significantly more progress could be made using tooling that currently exists or could be invented relatively easily.
>
>
> **W2: On negative results and impossibility perspectives.**
>
> We agree that impossibility results provide an important complementary lens and will revise the paper to explicitly discuss how negative results (e.g., in fairness and domain adaptation) can strengthen the predict–intervene–design framework by ruling out unproductive directions and clarifying fundamental limits.
>
> **Q1: On systematic generalization testing being “already standard.”**
>
> We agree that reviewers frequently request additional evaluations, including generalization checks. However, our point is more specific: current practice emphasizes post hoc robustness validation, rather than systematic, theory-driven generalization as a core object of study.
>
> Many findings are still derived from single models, single seeds, or fixed training setups, making it unclear whether they reflect fundamental properties or incidental artifacts (Sellam et al., 2022, Tigges et al., 2024, van der Wal et al., 2025). Even when additional evaluations are added during review, they are typically incremental extensions (e.g., more benchmarks or datasets), rather than deliberate attempts to identify invariances across training dynamics, data order, or model families.
>
> A key distinction is that a scientific approach requires explicitly defining the dimensions of variation over which a claim should generalize, and then systematically testing those dimensions. Without this, we cannot distinguish “regularities of the learning process” from “accidents of initialization”. Our recommendation is therefore not simply “do more evaluation,” but to reframe generalization as a central methodological principle tied to building predictive theories.
>
> **Q2: On scaling laws and empiricism vs. theory.**
>
> We agree scaling laws were discovered empirically and do not claim theory led this progress. We view them as a partial success case for the first stage of our hierarchy: prediction. They enable practitioners to predict training loss from model size and compute—exactly the kind of predictive capability we argue should extend to bias, memorization, and downstream behavior. At the same time, scaling laws illustrate the limitations of purely empirical discovery: they do not explain why these relationships hold or how to extend them to new settings. This gap motivates our broader agenda—moving from empirical regularities to causal, generalizable theories that support not just prediction but also intervention and design.

---

> > ### Author Rebuttal · Reviewer_Zi6w · 2026-04-02
> >
> > Thank you for your detailed response. I have no further questions, and I encourage you to incorporate these discussions into the next revised version. I would like to maintain my evaluation that this paper is a clear accept.

---

### Official Review · Reviewer_sNSo · 2026-03-13

**Significance:** 4
**Argument Clarity:** 4
**Rating:** 6
**Confidence:** 3

**Questions:**

Is it possible that mere “prediction” can also be considered as black-box and does not really explain why certain data or model characteristics will result in a certain outcome? How can this be mitigated?

**Alternative Views Section:**

Yes

**Compliance With Llm Reviewing Policy A Conservative:**

Affirmed.

**Discussion Potential:**

4

**Final Justification:**

After going through all the reviews and authors' response, I remain enthusiastic about the direction of future research this position paper may help steer. I will keep my original rating.

**Paper Summary:**

This paper presents the position that AI science should study the learning dynamics of models to establish casual models that can predict training outcome, allow interventions, and ultimately allow reverse design of training procedures with desired properties. To substantiate this position, the authors examined several research areas with both positive and negative evidence of progress, with which several research directions and priorities are identified.

**Position:**

Yes

**Position In Title:**

Yes

**Related Work:**

4

**Strengths And Weaknesses:**

Strengths

The proposed positioned to understand rather than describe, to move through the stages of prediction, intervention, and design utilizing the learning dynamics is well framed and insightful. It is highly relevant to the ICML community and has high discussion potential.

The position is substantiated by concrete and significant evidences, for both gaps and initial successes, across many areas. The evidence is based on substantial review of related works which provides solid support for the position proposed.

Clear open problems are identified and concrete recommendation for research directions and priorities are given which can fuel future research and discussions.


Weakness

None identified.

**Support:**

4

---

> ### Author Rebuttal · Authors · 2026-03-31
>
> **Q1: Is it possible that mere “prediction” can also be considered as black-box and does not really explain why certain data or model characteristics will result in a certain outcome? How can this be mitigated?**
>
> Yes, that is a real worry. The philosophy of science draws a distinction between a scientific law (an equation or regularity that makes accurate predictions) and a scientific theory (an account of the mechanisms that explain why that law holds). An illustrative historical example Kepler's laws of planetary motion predicted the orbits of the planets with remarkable accuracy decades before Newton provided any account of why those regularities held. The laws were useful, but it was Newton's theory of universal gravitation that told the first successful causal story that grounded them as consequences of something deeper.
>
> We believe that predictions are useful -- both intrinsically and as a way to validate scientific theories -- but our aim is to promote research that builds genuine scientific understanding through theories with explanatory power, not merely predictive accuracy.

---

> > ### Author Rebuttal · Reviewer_sNSo · 2026-04-04
> >
> > Thanks for the response. I think this is exactly what I was missing in the manuscript -- when discussing "prediction beyond description", it'd be good to incorporate discussion similar to the response to point out that even mere prediction, while as a step ahead of description, also has gap towards theory (which then can help bring out the other concepts). In the current manuscript, it may leave some audience (especially thinking about trainees) with the mis-concept that prediction is one of the end-goals: I guess it is, but it is more like a necessary condition, not sufficient. For a paper that is positioned to steer future research directions, I think it's important to make sure that such messages are probably conveyed to its readers.
> >
> > Regardless, I maintain my rating that this is a strong accept paper that everyone should read.

---

### Official Review · Reviewer_Thuf · 2026-03-13

**Significance:** 2
**Argument Clarity:** 2
**Rating:** 4
**Confidence:** 3

**Questions:**

See weaknesses.

**Alternative Views Section:**

Yes

**Compliance With Llm Reviewing Policy A Conservative:**

Affirmed.

**Discussion Potential:**

2

**Final Justification:**

After carefully reading the discussions between the authors and reviewers for this submission, I am now convinced that the results in this paper support the claim raised by the authors. However, in my view, even a position paper should include a clear and clean theoretical formulation of the claim, together with at least some experimental evidence to support its claims; strong theorems and extensive experiments are not necessary. Otherwise, the paper risks reading merely as a survey or a personal opinion piece. Nevertheless, I raise my score to 4.

**Paper Summary:**

This position paper suggests that AI research should move beyond treating trained models as static artifacts and instead study learning dynamics as the basis for a true science of AI. The authors propose a hierarchy of scientific maturity such as: predict, intervene, design, and suggest that progress in AI should be measured by our ability to forecast model properties from early signals, correct undesirable trajectories during training, and eventually design training procedures that reliably produce desired outcomes.

**Position:**

Yes

**Position In Title:**

Yes

**Related Work:**

3

**Strengths And Weaknesses:**

**Strengths**

- The paper raises an interesting issue. The critique of the current "fix it in post" mentality is clear and easy to understand, and the proposed predict-intervene-design framework is a useful organizing lens for thinking about scientific progress in AI.

- The paper is also broad in scope and connects several active areas such as: interpretability, bias, memorization, and learning dynamics, under one unifying theme. That gives the piece some value as a perspective article and may help motivate future research directions.

**Weaknesses**

- My main concern is that the paper does not provide any evidence for its central claims. There are no experiments, no formal theoretical results, and no concrete methodological contribution that would validate or sharpen the paper’s claims. Although this is a position paper, I still expect some good evidence for justify the claims.

- The authors repeatedly argue that AI should become predictive, interventionist, and design-driven, but the paper does not demonstrate how to get there beyond high-level desiderata. As a result, the contribution feels more like a manifesto than a paper with actionable technical content.

- The general call for studying training dynamics, making stronger predictions, and seeking causal understanding is important, but these themes are already widely discussed across scaling-law work, mechanistic interpretability, memorization studies, and alignment research. The paper frames them clearly, but does not sufficiently distinguish its contribution from an accessible synthesis of ongoing conversations.

**Support:**

2

---

> ### Author Rebuttal · Authors · 2026-03-31
>
> Thank you for your feedback. We respond to each point below and have several questions we hope will help us understand your concerns more precisely and improve the paper.
>
> **W1: Evidence for our claims**
>
> Our primary claim, about how to do good scientific research, is a philosophical one not an empirical one. While earlier drafts contained more detailed philosophical arguments grounded in the philosophy of science literature, we felt it would be more useful to most readers to go into more detail on examples of work we felt did a good job or were oriented in a good direction. Doing a whole empirical research project feels out of scope for a position paper. That’s why we ground our ideas in previous work. But we would love to update the paper to be more compelling in your eyes. Can you provide more detail about what you would find compelling?
>
> **W2: The contribution feels more like a manifesto than a paper with actionable technical content.**
>
> Our primary aim with this paper is to convince researchers to adopt our view of what it looks like to do high quality scientific research. However, actually implementing that is necessarily problem-specific. We agree that a common issue in papers like this is that someone might read it and go “okay, I agree that the path of research should change but what should I actually do.” That’s why we’ve tried to ground our discussion in the best work in this vein we are aware of and highlight both successes and important open problems. Does the reviewer feel that those examples don’t provide adequate guidance? What would it look like in your view to do better at this?
>
> **W3: The general call for studying training dynamics, making stronger predictions, and seeking causal understanding is important, but these themes are already widely discussed across scaling-law work, mechanistic interpretability, memorization studies, and alignment research. The paper frames them clearly, but does not sufficiently distinguish its contribution from an accessible synthesis of ongoing conversations.**
>
> The reviewer is correct to note that these topics are discussed by some in various communities, however that's very different from saying that the approach we advocate is practiced or that the position that we advocate for is mainstream. Often times this view of research is not even recognized by the community. For example, NeurIPS 2025's MechInterp workshop held a debate about different views of interpretability research with no speakers representing the views articulated in this paper.
>
> Several of the papers we cite make it clear how unusual work like this is. Tigges et al. (2024) observes that despite the fact that circuit research is very common in mechanistic interpretability, the vast majority of papers look only at a single model or a small number of models and that none of the hundreds of papers published over the preceding several years look at the learning dynamics of circuits in non-toy models. Sellam et al. (2022) document that BERT experiments are overwhelmingly based on single checkpoints and show that re-running pretraining leads to substantially different performance, motivating MultiBERTs as a resource precisely because the field lacked the infrastructure to make principled statements about procedures. Similar concerns motivate van der Wal et al. (2025).
>
> Another illustrative example is controlling social bias. There have been hundreds if not thousands of *descriptive* papers on social biases in LLMs, but despite the fact that such papers almost universally recognize social bias as a direct consequence of representation within the training data there is virtually no research on interventions to control gender bias at that level. The only such work we are familiar with is a demo in Biderman et al. (2024b) where they use this problem as an example of the type of problem that Pythia could help enable people to tackle. Despite the Pythia paper receiving around 2,000 citations, we were unable to find any work picking up that explicit challenge among the papers citing Pythia or subsequent suitable models suites such as OLMo. The closest we were able to find are "Fairness Dynamics During Training," which descriptively (rather than prescriptively) analyzes the evolution of gender bias in Pythia models and "The Bias Amplification Paradox in Text-to-Image Generation," which provides compelling evidence that bias amplification is overstated due to methodological issues in prior work but doesn't not address the question of predicting or controlling biases deliberately.
>
> At its core, we believe the reviewer's objections boil down to "everyone knows this." Respectfully, we disagree. Based on our experience in these communities we believe that some of the ideas we are advocating for are extremely niche in the fields the reviewer mentions and that even among people who agree with us philosophically, many don't do the research we are advocating for.

---

> > ### Author Rebuttal · Reviewer_Thuf · 2026-04-04
> >
> > I am reading the authors’ responses as well as the comments from the other reviewers, and I will write my response later.

---

### Decision · Program_Chairs · 2026-04-30

**Decision:**

Accept (spotlight)

**Comment:**

Reviewers unanimously praised the work for its breadth of thought, clarity, inclusion of open problems, and timeliness. A few weaknesses were identified that the rebuttals largely resolved.

This is a clear accept.